# Towards developing a beef meat export oriented policy in Tanzania: -Exploring the factors that influence beef meat exports-

**Cornel Anyisile Kibona**[1,2]*, **Zhang Yuejie**[1], **Lu Tian**[1]

**1** College of Economics and Management, Jilin Agricultural University, Changchun, Jilin, China,
**2** Department of Agricultural Economics and Finance, Mwalimu Julius. K. Nyerere University of Agriculture and Technology, Musoma, Tanzania

* cornelkibona@gmail.com

## Abstract

The purpose of this study was to investigate the factors that influence beef meat exports in Tanzania, with a particular focus on the years 1985 to 2020, in enhancing the development of beef meat export-oriented policy in Tanzania, thereby enhancing beef exports in Tanzania. A time-series panel dataset was analyzed using both descriptive statistics and ordinary least squares (OLS) linear regression analyses models. As per the descriptive analyses, beef meat exports reached the highest pick of 4,300 tons per year in 1990, whereas from 1991 to date, beef meat exportation in Tanzania has been in declining trends despite an increase in beef meat output and trade openness from 162,500 to 486,736 tons and 7.6 to 98.7%, respectively. Nevertheless, while the prospect of Tanzanian beef meat exports appears bright and promising, the industry will continue to encounter trade barriers and must stay competitive to produce enough volume and quality beef meat to meet the needs of its existing and expanding markets. This is because, Tanzanian beef meat competes for market share with beef meat from other countries in the global markets, where customers pay a premium for lines of beef meat that meet quality standards while discarding those that do not. This indicates that the quantity of beef meat produced has no relevance to its world market share, but its quality standards do. Furthermore, the econometric results revealed that the coefficients of the terms of trade, Tanzania GDP per capita, global beef meat consumption, trade openness, and beef meat outputs were found to be significantly positive ($P < 0.05$) influencing beef meat exports in Tanzania, whereas the trading partners' GDP per capita and exchange rate were not. The findings have varying implications as to what factors need to be addressed to further improve beef meat exports. From the farmer's perspective, better access to adequate funds as a result of increased income benefit from export will assist in improving beef cattle productivity and quality to compete effectively in the global markets. From the government's perspective, because trade openness promotes economic growth through export benefits, the Tanzania government and policymakers need to establish balanced policies to strengthen bilateral trade relationships to generate more opportunities in global markets.

**Data Availability Statement:** All relevant data are within the article and its Supporting Information files.

**Funding:** This research is supported and funded by the National Beef Cattle Industrial Technology System and Industrial Economy Research Project under the Ministry of Agriculture in the People's Republic of China (PRC) (CARS-37). The funders had no role in study design, data collection and analysis, decision to publish, or preparation of the manuscript.

**Competing interests:** The authors have declared that no competing interests exist.

## 1. Introduction

The protracted forecast for worldwide consumption of beef meat is generally optimistic, supported by increasing population and rising family prosperity (economic growth in developing markets [1]. From 2007 to 2018, the market price rose at an average yearly rate of +3.2%; the trend pattern remained stable, with some noteworthy swings during the investigated period [2]. The highest rate of growth was achieved in 2008, with a rise of 11% year on year. Global beef meat consumption peaked in 2018 and is anticipated to rise further in the coming years [2]. Moreover, worldwide beef meat demand is anticipated to boost at a 0.6% annual rate throughout 2024 [3]. This increase is estimated to result in an incremental 1.8 million tons of beef meat consumed globally by 2024 [3]. This statistic shows the overall opportunities for exporters including Tanzania.

Worldwide beef meat output (production) was expected to shrink by 2% in 2020, owing mostly to the impact of the COVID-19 crisis. However, the United States Department of Agriculture (USDA) forecasts worldwide beef meat production quantities to rise to 61.5 million tonnes of carcass weight equivalents (CWE) through 2023, a 2% increase over 2020 rates. This is simply because; the worldwide economy is likely to promote beef meat industry development, particularly in countries affected by COVID-19 setbacks [1]. Besides that, the vast majority of states desire to minimize their trade imbalances through importing [4]. Most importantly, the largest global beef meat exporters, such as Brazil, Argentina, and Australia, are facing transportation issues in supplying their beef meat to the global markets [4]. Because of faster growth in exports from developing countries, beef meat trade growth for developed countries will slow to an annual average rate of 1.4% by 2028, compared to 3% during the previous decade [5]. As a result, developing countries like Tanzania can benefit if they take the effort to grasp the chance [4].

Tanzania is one of the leading African countries in terms of beef cattle population, next only to Ethiopia and Sudan [6]. According to the Ministry of Livestock and Fisheries census report 2020, the country had over 34.5 million cattle in 2020/21, an increase from 33.4 million in 2019/20 [6]. Despite being third in Africa in terms of cattle population, the beef cattle industry contributes just 5.9% of Tanzania's GDP, and the growth rate is minimal at 2.8% [7]. In addition, the country has yet to completely take positive action in beef meat exports [6]. Most beef meat exports to international markets (trading partners) are to Kenya, Uganda, Dubai, Iraq, Comoro, Oman, Malaysia, Hong Kong, Vietnam, etc [8]. The amount of beef meat exported is minimal in comparison to the potential because the foreign market demands higher quality standards, which is what most local farmers are unable to achieve practically. Thus, there is a potential to improve efficient and competitive beef meat production to fulfil domestic and global market criteria. The scale of this market is comparable to the quantity of beef meat imported each year, which is approximately 700 tons [9, 10].

Hence, the Tanzania government has been emphasizing modernization of the beef meat industry and reinforcing its initiatives to maintain adherence to hygienic prerequisites for global trade to re-establish abandoned international beef meat markets such as Mauritius and attain access to advanced global export markets [7]. The invested US$ 596 million in the Tanzania livestock master plan (TLMP) is bound to enhance the industry's effectiveness and eradicate poverty, boosting the country's GDP, increasing food security and nutrition, and creating new jobs [7, 10]. The modernization of the beef meat industry has a significant impact on attaining the national development vision 2025 as well as the national strategy for economic and poverty reduction (NSPR) through beef meat exports [4]. Furthermore, the European Union (EU)-funded STOSAR project "support towards effective implementation of the SADC Regional Agricultural Policy," which has been executed by FAO, is assisting in the protracted

capacity building of industry service providers, farmers, and supply network drivers, notably in enhancing animal health facilities to improve disease defense systems [7]. Generally, the project intends to enhance long-term advancement in the commercial beef meat industry by improving access to domestic and international beef meat markets and facilitating bilateral commercial relations [7].

As a result, meat output grew by 5.2% reaching 738,166 tons in 2020/21, slightly higher compared to 701,679.1 tons in 2019 [6]. About 508,000 tons (69%) of the total 738,166 tons of meat were produced from beef cattle. Notwithstanding, only a tiny amount of beef meat is being exported [6]. This is an indication of the low efficiency of beef meat exports in Tanzania. It is a point of anxiety for Tanzania since the economy is heavily reliant on the foreign sector. In general, a country's export sector is determined by a mixture of aspects such as supply-demand concerns [11], trade relations expenditures [12], capital inflows (economic growth), and trade facilitation tactics (access to profitable markets) [13]. According to statistics, although beef meat demand in Tanzania is currently at 53% [14], it is anticipated that the international markets will expand by 13.7% [15]. Tanzania's export potential will grow as a result; therefore, the government had to design and implement effective strategies (policies) to fully maximize the potential, as rising demand is expected to be met by developing countries like Tanzania.

## 2. A brief literature review on the importance of beef meat exports and the beef meat value chain

### 2.1. Importance of beef meat exports

The beef meat industry has the opportunity to profoundly improve the economy by supplying beef meat to the emerging international markets. According to the Peel analysis [16], beef meat exports add value by increasing the volume of total beef meat sales, allowing producers (exporters) to sell more beef meat to more places. Furthermore, beef meat exports increase the value of beef meat by selling it at higher prices. However, this is commonly owing to the reasons that beef meats with a remarkably low value in the producer countries (exporters) have a higher value in some international markets [16, 17]. Besides that, beef meat exports add value by maximizing the diverse set of beef meat products available in the domestic beef meat market [16]. Beef meat demand is commonly classified by pork and poultry as the primary substitutes for beef. In reality, the most prevalent substitute for any given beef meat product is another beef meat product [16]. Further to that, beef meat export markets provide a way to ship out some of those lower valued cuts thereby focusing domestic demand on higher-value beef. More importantly, the role of beef meat exports in improving domestic product mix and optimizing beef meat demand is generally ignored, but it is essential in terms of total beef meat export value [17].

In terms of the prospective macroeconomic and employment effects of increased beef meat exports, the results of a social accounting matrix (SAM) analysis by [18] show that higher export demand for beef meat generates more GDP and household income than a similar shock in live beef cattle export demand. Increased beef meat exports, for example, increase GDP by 0.09% more and household income by 0.06% more than increased live beef cattle exports. The value of production output, on the other hand, rises significantly more (by 0.3 percent) under a particular circumstance of increased beef meat export, apparently due to higher efficiencies for beef meat as compared to live beef cattle. Likewise, higher beef meat exports resulted in higher employment.

### 2.2. Beef meat value chain in Tanzania

The value chain describes the sequence of activities needed to move a commodity from its initial production point to its final point of consumption [19, 20]. In Tanzania, the beef meat

value chain consists of live beef cattle, fresh beef meat, processed beef meat products, and by-products that are sold both domestically and on the international market [20]. Actors in the chain include primary producers, traders in live beef cattle, beef meat, and by-products, processors, butchers, other retail outlets, and consumers [20]. The majority of actors are not specialized, and their functions are related to different sections of the value chain. Numerous primary producers, for example, participate in animal trading, and some upstream actors, such as butchers, trade in live beef cattle and beef meat and invest in primary processing to produce higher-value cuts, minced beef meat, and sausages [20]. Many technical and institutional impediments impede the supply and use of inputs, as well as production and processing, marketing, and retailing. The chain is disjointed, disorganized, uncontrolled (despite being heavily regulated), and poorly coordinated [21]. It is occupied by a multitude of small stockholders [22], an unidentified but undisputedly enormous number of intermediaries who function across every link, and a correspondingly unidentified number of small processors and butchers who put products on the market for the consumer but primarily lack the technical and financial ability to run it effectively and productively. The value chain's horizontal and vertical linkages are generally weak and non—competitive, and they require assistance to be strengthened [23]. It is necessary to have strictly enforced and articulated standards, as well as a legal regulatory framework. In Tanzania, most of these necessities are still weak, non-existent, or are not enforced [20].

Given the significance of the beef meat industry, it is critical to investigate the essential factors influencing beef meat exports to its trading partners towards developing a beef meat exports-oriented policy in Tanzania. Mostly to the best of the authors' knowledge, there is almost little or no estimate of the effectiveness of Tanzanian beef meat exports in the existing studies.

Generally, the research question is from 1985 to 2020, how has beef meat output (beef meat production), exchange rates, Tanzania's gross domestic product (GDP) per capita, gross domestic product (GDP) per capita of trading partners, exchange rate, global beef meat consumption, trade openness, and country's terms of trade influenced Tanzania's beef meat exports?. The findings will aid in addressing beef meat export failure problems in Tanzania's beef meat industry, allowing for more appropriate actions to improve beef meat exports. Additionally, this is critical for developing a successful development set of policies for optimizing economically efficient beef meat supply to worldwide markets, hence boosting wealth creation in the beef meat industry [24].

## 3. Methodology

### 3.1. Dataset and sources

This study makes use of a time series panel dataset of Tanzania's international beef meat exports to her trading partners (export markets) from 1985 to 2020. Data has been collected at the highest feasible consistency. Table 1 illustrates the data sources as well as descriptions of the variables, whereas Table 2 provides the original dataset.

### 3.2. Ethical considerations

The Jilin Agricultural University Graduate Research Ethics Committee in China authorized this work. The Tanzania Ministry of Livestock and Fisheries (MLF) then authorized it with the reference number (AB.16/2020/01). Because the data collecting procedure used time-series panel data collection, no human participants were engaged; hence, consent was not acquired.

**Table 1. Sources of data and descriptions of variables.**

| Variables | Descriptions of Data | Sources of Data |
|---|---|---|
| Beef meat export quantity at market price | The total amount of beef meat exported to trading partners. | ▪ FAOSTAT [25],<br>▪ UNCOMTRADE database [26]. |
| Terms of trade (TOT)-Tanzania | A ratio of import prices to export prices (Index of export prices/index of import prices)*100. | ▪ FAOSTAT [25],<br>▪ COMTRADE database [26],<br>▪ Author's calculation. |
| Real exchange rate | Tanzania shillings per US dollar divided by the importing country's currency per US dollar (based on purchasing power parity-PPP). | ▪ FAOSTAT [25]. |
| Tanzania GDP per capita (US$) | Gross Domestic Product at a market price based on purchasing power parity (PPP). | ▪ FAOSTAT [25]. |
| GDP per capita of trading partners(US$) | Gross Domestic Product of the importers at a market price based on purchasing power parity (PPP). | ▪ FAOSTAT [25],<br>▪ Author's calculation. |
| Global beef meat consumption | Total beef meat consumed globally is measured in thousand tonnes of carcass weight. | ▪ Organization for Economic Co-operation and Development (OECD) (2022) [27]. |
| Trade openness (TO-%) | The value of merchandise trade (exports plus imports) is a percent of gross domestic product (GDP). Is a measure of the extent to which a country is engaged in the global trading system. | ▪ World integrated trade solutions (WITS) [28]. |
| Beef meat production quantity | Total beef meat output produced in Tanzania. | ▪ FAOSTAT [25]. |

## 3.3. Conceptual framework

To address beef meat export failure issues in Tanzania's beef meat industry, enabling more effective initiatives to improve beef meat exports, thereby boosting wealth generation in the beef meat industry. This study hypothesized that the enhanced performance of the Tanzanian beef meat exports is reliant on the link between increased beef meat export and its internal and external drivers (factors). The external variable factors include GDP per capita of trading partners, global beef meat consummation, and trade openness, while internal variable factors include terms of trade, exchange rates, Tanzania GDP per capita, and beef meat output (production). Furthermore, once these internal and external factors have been thoroughly investigated, they open the door to effective policy formulation for improving economically efficient beef meat supply to global markets (exportation policy), thereby increasing Tanzania's beef meat exports. The increased beef meat exports then enable wealth generation in the beef meat industry, which promotes further country economic growth. Fig 1 depicts the interrelations of the influential variables of beef meat exports in this study and how they are interlinked.

## 3.4. Data analysis models

To analyze data, this study utilized descriptive and inferential statistics, as well as regression (econometric) analyses models.

Descriptive and inferential statistics involved mean and percentages of beef meat exports quantity ($Y_1$), terms of trade ($X_1$), exchange rates ($X_2$), Tanzania GDP per capita ($X_3$), GDP per capita of trading partners ($X_4$), global beef meat consumption ($X_5$), trade openness ($X_6$) and beef meat production quantity ($X_7$) for the period from 1985 to 2020, presented in the table and line graphs.

Regression (econometric) analysis; to analyze the time series data for this study, the ordinary least squares (OLS) estimation approach was used. The ordinary least squares (OLS) multiple linear regression model was applied to investigate the factors (drivers) that especially influenced beef meat exports quantity to trading partners (global markets). The Ordinary Least Squares (OLS) model was chosen because it is a numerical (mathematical) modeling tool that can be used to explain (describe) the relationship between a continuous dependent

**Table 2. Original dataset.**

| Year | Beef meat exports | Terms of trade | Exchange rates | Tanzania GDP per capita | GDP per capita of Trading partners | Global beef meat consumption | Trade openness | Beef Meat output |
|------|------|------|------|------|------|------|------|------|
| | (Tons) | (%) | (US$) | (US$) | (US$) | (Tons) | (%) | (Tons) |
| | $Y_1$ | $X_1$ | $X_2$ | $X_3$ | $X_4$ | $X_5$ | $X_6$ | $X_7$ |
| 1985 | 0 | 0.0 | 17.5 | 511.9 | 5,979.6 | 43,001,790 | 7.6 | 162,500 |
| 1986 | 92 | 82.2 | 32.7 | 350.9 | 4,977.8 | 44,534,056 | 8.2 | 170,200 |
| 1987 | 0 | 0.0 | 64.3 | 238.6 | 5,305.2 | 45,735,464 | 9.7 | 171,400 |
| 1988 | 0 | 0.0 | 99.3 | 269.9 | 5,253.5 | 46,990,825 | 10.4 | 185,900 |
| 1989 | 0 | 0.0 | 143.4 | 279.5 | 5,671.0 | 47,260,338 | 11.7 | 193,600 |
| 1990 | 4,300 | 3,888.9 | 195.1 | 264.2 | 6,058.0 | 48,001,790 | 12.8 | 195,200 |
| 1991 | 0 | 0.0 | 219.2 | 288.8 | 5,648.4 | 49,534,056 | 13.1 | 200,000 |
| 1992 | 0 | 0.0 | 297.7 | 249.1 | 6,348.7 | 53,735,464 | 14.6 | 205,000 |
| 1993 | 0 | 0.0 | 405.3 | 220.1 | 6,590.9 | 53,990,825 | 15.5 | 210,000 |
| 1994 | 0 | 0.0 | 509.6 | 219.9 | 6,921.2 | 55,260,338 | 16.3 | 213,000 |
| 1995 | 0 | 0.0 | 574.8 | 249.1 | 7,402.0 | 55,983,297 | 17.7 | 246,000 |
| 1996 | 0 | 0.0 | 580.0 | 300.2 | 8,082.9 | 56,277,813 | 18.5 | 194,000 |
| 1997 | 81 | 191.7 | 612.1 | 346.3 | 8,513.5 | 57,214,880 | 19.5 | 193,000 |
| 1998 | 122 | 142.0 | 664.7 | 368.4 | 7,627.7 | 57,299,196 | 20.8 | 198,000 |
| 1999 | 231 | 710.3 | 744.8 | 368.9 | 8,097.5 | 58,130,253 | 21.6 | 260,000 |
| 2000 | 20 | 40.6 | 800.4 | 378.1 | 9,520.1 | 58,775,737 | 22.2 | 230,000 |
| 2001 | 24 | 51.9 | 876.4 | 375.9 | 9,033.0 | 57,627,333 | 24.7 | 181,000 |
| 2002 | 1 | 0.6 | 966.6 | 380.7 | 9,239.1 | 59,808,478 | 24.4 | 182,000 |
| 2003 | 2 | 1.2 | 1,038.4 | 399.1 | 10,138.0 | 60,689,037 | 28.7 | 182,500 |
| 2004 | 3 | 1.2 | 1,089.3 | 427.3 | 11,643.6 | 61,416,191 | 32.8 | 184,000 |
| 2005 | 46 | 12.8 | 1,128.9 | 458.0 | 13,689.1 | 62,534,418 | 35.1 | 204,520 |
| 2006 | 14 | 12.3 | 1,251.9 | 490.2 | 15,452.7 | 64,599,699 | 42.6 | 208,046 |
| 2007 | 0 | 0.7 | 1,245.0 | 550.7 | 16,619.8 | 66,169,507 | 44.4 | 180,629 |
| 2008 | 113 | 39.0 | 1,196.3 | 681.3 | 19,437.4 | 65,333,043 | 52.3 | 218,976 |
| 2009 | 23 | 20.9 | 1,320.3 | 690.5 | 15,026.3 | 64,826,857 | 42.7 | 225,178 |
| 2010 | 15 | 1.0 | 1,409.3 | 730.0 | 16,362.0 | 65,015,786 | 50.5 | 243,943 |
| 2011 | 0 | 0.0 | 1,572.1 | 764.4 | 19,349.6 | 64,568,177 | 63.5 | 262,606 |
| 2012 | 1,673 | 34.7 | 1,583.0 | 858.2 | 20,275.3 | 65,185,452 | 58.8 | 289,835 |
| 2013 | 3 | 0.1 | 1,600.4 | 964.2 | 20,274.6 | 66,372,181 | 61.3 | 299,581 |
| 2014 | 35 | 1.6 | 1,654.0 | 1,025.6 | 19,974.1 | 66,024,414 | 67.7 | 309,353 |
| 2015 | 37 | 1.4 | 1,991.4 | 943.3 | 16,580.7 | 66,223,268 | 70.4 | 319,112 |
| 2016 | 429 | 58.8 | 2,177.1 | 962.0 | 15,873.4 | 67,505,862 | 71.6 | 323,775 |
| 2017 | 1,155 | 152.1 | 2,228.9 | 1,001.2 | 16,978.5 | 68,461,965 | 80.7 | 394,604 |
| 2018 | 324 | 30.3 | 2,263.8 | 1,041.0 | 18,598.1 | 69,499,688 | 89.3 | 471,692 |
| 2019 | 180 | 0.0 | 2,288.2 | 1,085.1 | 18,026.8 | 70,474,255 | 93.9 | 506,798 |
| 2020 | 955 | 101.9 | 2,294.1 | 1,124.3 | 19,557.8 | 70,881,971 | 98.7 | 486,736 |

Sources: FAOSTAT [25], UNCOMTRADE database [26], Organization for Economic Co-operation and Development (OECD) (2022) [27], World integrated trade solutions (WITS) [28] and, Author's calculation.

variable (the amount of beef meat exported) and several independent variables [29] as cited by Kibona and Yuejie [30]. Moreover, the OLS allows for co-integration and stationary checks to be performed.

Beef meat exports quantity ($Y_1$) was hypothesized as the function of the terms of trade ($X_1$), exchange rates ($X_2$), Tanzania GDP per capita ($X_3$), GDP per capita of trading partners ($X_4$),

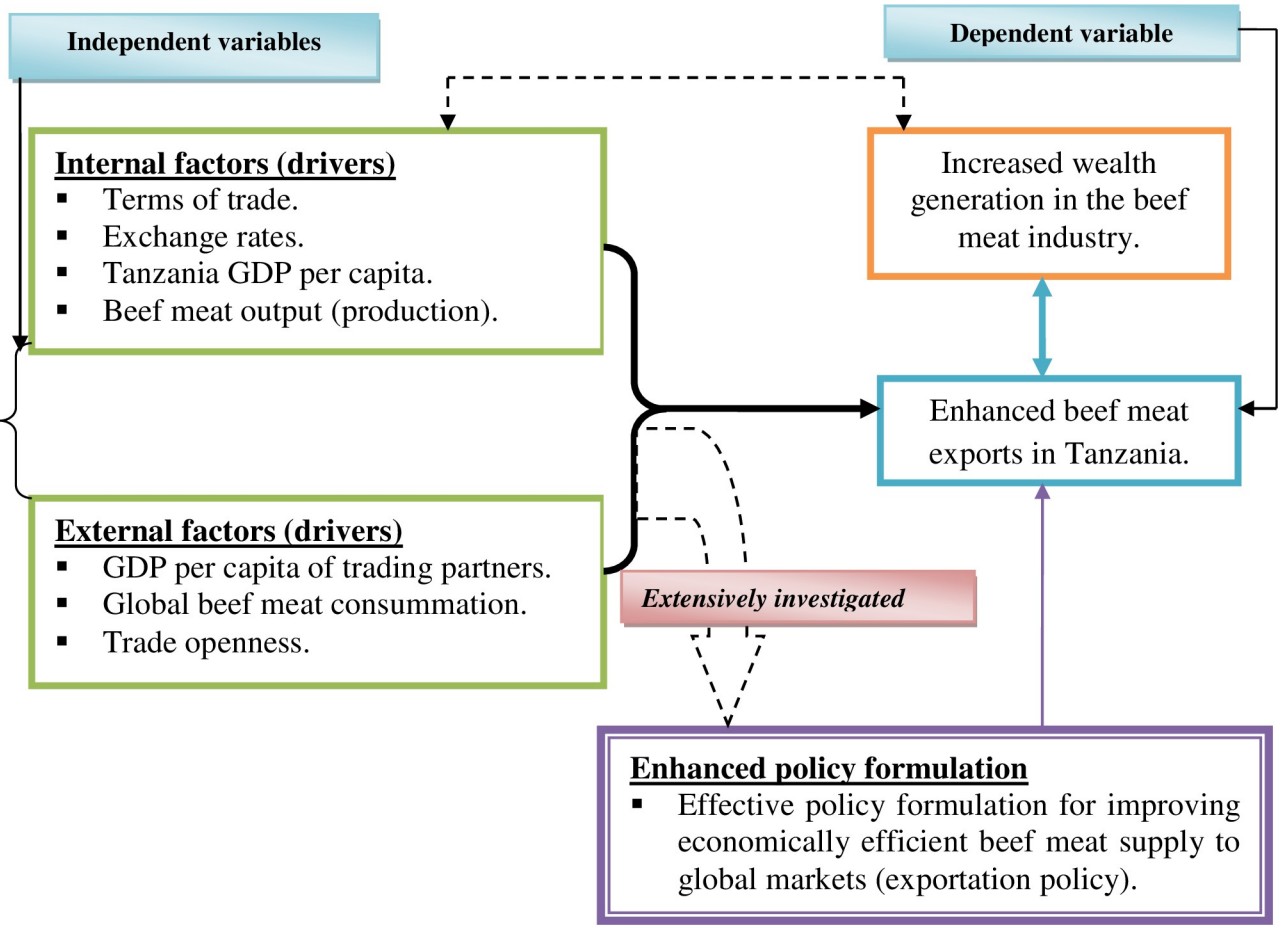

**Fig 1. Conceptual framework.**

global beef meat consumption ($X_5$), trade openness ($X_6$), and beef meat production quantity (beef meat output) ($X_7$) (see Eq 1).

$$Beef\ Meat\ Exports\ Quantity = f(x_1,\ x_2, x_3, x_4, x_5,\ x_6,\ x_7) \qquad (1)$$

Thus, the actual OLS model for beef meat exports in Tanzania is expressed as:

$$Y_i = \beta_0 + \beta_1 X_1 + \beta_2 X_2 + \beta_3 X_3 + \ \ldots\ldots.. + \beta_n X_n + \varepsilon_i \qquad (2)$$

Here;

$Y_i$ represents the amount of beef meat exported, $\beta_0$ is a constant (intercept), $\beta_1, \ldots.,$ and $\beta_n$ are the coefficients to be estimated and $X_1, \ldots.,$ and $X_n$ represent the vectors of the independent variables (drivers of beef meat exports in Tanzania), $f$ indicates the function of, and $\varepsilon_i$ is the error term.

To estimate the elasticity of the beef meat export variable relative to independent variables, the authors first normalized the regression equation model shown above employing a natural logarithm. A natural logarithm eliminates abnormalities by converting the non-linear

regression equation into a linear regression equation [4]. Thus, the derived equation is simplified to;

$$LnY_{it} = \beta_0 + \beta_1 Lnx_{1t} + \beta_2 Lnx_{2t} + \beta_3 Lnx_{3t} + \beta_4 Lnx_{4t} + \beta_5 Lnx_{5t} + \beta_6 Lnx_{6t} + \beta_7 Lnx_{7t} + \varepsilon_t \quad (3)$$

Here;

$LnY_{it}$ = represents the among of beef meat exported in natural logarithm,

$Ln$X$_1$ = terms of trade in natural logarithm

$Ln$X$_2$ = exchange rates in natural logarithm

$Ln$X$_3$ = Tanzania GDP per capita in natural logarithm

$Ln$X$_4$ = GDP per capital of trading partners in natural logarithm

$Ln$X$_5$ = Global beef meat consumption in natural logarithm

$Ln$X$_6$ = Trade openness in natural logarithm

$Ln$X$_7$ = Beef meat production quantity in natural logarithm

t = 1985 . . . 2020 yearly series

$Ln$ = stands for natural logarithm.

Hence, the compressed ordinary least squares (OLS) regression model for drivers (factors) influencing beef meat export quantity in Tanzania was defined as follows (see Eq 4):

$$
\begin{aligned}
LnBeef\ Meat\ Export\ Quantity \\
= \beta_0 + \beta_1 LnTerms\ of\ trade + \beta_2 LnExchange\ rate \\
+ \beta_3 LnTanzania\ GDP\ per\ capita \\
+ \beta_4 LnGDP\ per\ capital\ of\ trading\ partners \\
+ \beta_5 LnGlobal\ beef\ meat\ consumption + \beta_6 LnTrade\ openness \\
+ \beta_7 LnBeef\ meat\ production\ quantity (Beefmeat\ output) \\
+ \varepsilon
\end{aligned}
\quad (4)
$$

Hence, the converted dataset in natural logarithm (*Ln*) is shown in Table 3.

## 4. Results and discussions

### 4.1. Descriptive results

**4.1.1. The general changes and relationships between beef meat exports, beef meat outputs, and trade openness in Tanzania from 1985 to 2020.** Fig 2 depicts the trends of beef meat exports in Tanzania from 1985 to 2020. While Table 4 shows the general changes and relationships between beef meat exported, beef meat output, and trade openness from 1985 to 2020. Generally, Fig 2 illustrates that the maximum amount of beef meat exported was 4,300 tons in 1990, with no beef meat exported from 1991 to 1997. From 1998 to 2011, there are both positive and negative changes. The export amount increased to 1673 tons in 2012, decreased to 3 tons in 2013, and then increased again, from 35 tons in 2014 to 1155 tons in 2017. From 2018 to 2019, there was a declining trend. Again the rising trend emerged in 2020. Such little exportation is raising the question of why despite the improving trends of beef meat outputs and trade openness as shown in Table 4, the exportation remained low which resulted in a 0.2% increase in the proportion of beef meat exported to total beef meat outputs.

**Table 3. Dataset converted into natural logarithm (*Ln*).**

| Year | Beef meat exports | Terms of trade | Exchange rates | Tanzania GDP per capita | GDP per capita of Trading partners | Global beef meat consummation | Trade openness | Beef Meat output |
|---|---|---|---|---|---|---|---|---|
| | (Tons) | (%) | (US$) | (US$) | (US$) | (Tons) | (%) | (Tons) |
| | $LnY_1$ | $LnX_1$ | $LnX_2$ | $LnX_3$ | $LnX_4$ | $LnX_5$ | $LnX_6$ | $LnX_7$ |
| **1985** | **0.0** | **0.0** | **2.9** | **6.2** | **8.7** | **17.6** | **2.0** | **12.0** |
| 1986 | 4.5 | 4.4 | 3.5 | 5.9 | 8.5 | 17.6 | 2.1 | 12.0 |
| 1987 | 0.0 | 0.0 | 4.2 | 5.5 | 8.6 | 17.6 | 2.3 | 12.1 |
| 1988 | 0.0 | 0.0 | 4.6 | 5.6 | 8.6 | 17.7 | 2.3 | 12.1 |
| 1989 | 0.0 | 0.0 | 5.0 | 5.6 | 8.6 | 17.7 | 2.5 | 12.2 |
| 1990 | 8.4 | 8.3 | 5.3 | 5.6 | 8.7 | 17.7 | 2.5 | 12.2 |
| 1991 | 0.0 | 0.0 | 5.4 | 5.7 | 8.6 | 17.7 | 2.6 | 12.2 |
| 1992 | 0.0 | 0.0 | 5.7 | 5.5 | 8.8 | 17.8 | 2.7 | 12.2 |
| 1993 | 0.0 | 0.0 | 6.0 | 5.4 | 8.8 | 17.8 | 2.7 | 12.3 |
| 1994 | 0.0 | 0.0 | 6.2 | 5.4 | 8.8 | 17.8 | 2.8 | 12.3 |
| 1995 | 0.0 | 0.0 | 6.4 | 5.5 | 8.9 | 17.8 | 2.9 | 12.4 |
| 1996 | 0.0 | 0.0 | 6.4 | 5.7 | 9.0 | 17.8 | 2.9 | 12.2 |
| 1997 | 4.4 | 5.3 | 6.4 | 5.8 | 9.0 | 17.9 | 3.0 | 12.2 |
| 1998 | 4.8 | 5.0 | 6.5 | 5.9 | 8.9 | 17.9 | 3.0 | 12.2 |
| 1999 | 5.4 | 6.6 | 6.6 | 5.9 | 9.0 | 17.9 | 3.1 | 12.5 |
| 2000 | 3.0 | 3.7 | 6.7 | 5.9 | 9.2 | 17.9 | 3.1 | 12.3 |
| 2001 | 3.2 | 3.9 | 6.8 | 5.9 | 9.1 | 17.9 | 3.2 | 12.1 |
| 2002 | 0.0 | -0.6 | 6.9 | 5.9 | 9.1 | 17.9 | 3.2 | 12.1 |
| 2003 | 0.7 | 0.2 | 6.9 | 6.0 | 9.2 | 17.9 | 3.4 | 12.1 |
| 2004 | 1.1 | 0.2 | 7.0 | 6.1 | 9.4 | 17.9 | 3.5 | 12.1 |
| 2005 | 3.8 | 2.5 | 7.0 | 6.1 | 9.5 | 18.0 | 3.6 | 12.2 |
| 2006 | 2.6 | 2.5 | 7.1 | 6.2 | 9.6 | 18.0 | 3.8 | 12.2 |
| 2007 | 0.0 | -0.4 | 7.1 | 6.3 | 9.7 | 18.0 | 3.8 | 12.1 |
| 2008 | 4.7 | 3.7 | 7.1 | 6.5 | 9.9 | 18.0 | 4.0 | 12.3 |
| 2009 | 3.1 | 3.0 | 7.2 | 6.5 | 9.6 | 18.0 | 3.8 | 12.3 |
| 2010 | 2.7 | -0.0 | 7.3 | 6.6 | 9.7 | 18.0 | 3.9 | 12.4 |
| 2011 | 0.0 | 0.0 | 7.4 | 6.6 | 9.9 | 18.0 | 4.2 | 12.5 |
| 2012 | 7.4 | 3.5 | 7.4 | 6.8 | 9.9 | 18.0 | 4.1 | 12.6 |
| 2013 | 1.1 | -2.4 | 7.4 | 6.9 | 9.9 | 18.0 | 4.1 | 12.6 |
| 2014 | 3.6 | 0.5 | 7.4 | 6.9 | 9.9 | 18.0 | 4.2 | 12.6 |
| 2015 | 3.6 | 0.3 | 7.6 | 6.8 | 9.7 | 18.0 | 4.3 | 12.7 |
| 2016 | 6.1 | 4.1 | 7.7 | 6.9 | 9.7 | 18.0 | 4.3 | 12.7 |
| 2017 | 7.1 | 5.0 | 7.7 | 6.9 | 9.7 | 18.0 | 4.4 | 12.9 |
| 2018 | 5.8 | 3.4 | 7.7 | 6.9 | 9.8 | 18.1 | 4.5 | 13.1 |
| 2019 | 5.2 | 0.0 | 7.7 | 7.0 | 9.8 | 18.1 | 4.5 | 13.1 |
| 2020 | 6.9 | 4.6 | 7.7 | 7.0 | 9.9 | 18.1 | 4.6 | 13.1 |

Certainly, more trade liberalization (trade openness) allows for increased global trade and market connectivity. The analysis in Table 4 shows that trade openness has increased rapidly from 7.6% in 1985 to 98.7% in 2020, representing a 91.1% increase in free (minimal export restrictions) exports to trading partners. This increase suggests that beef meat exports in Tanzania could increase proportionally, though exportation remains minimal. This could be because Tanzanian beef meat competes for market share with beef meat from other countries in global beef meat markets. As a result, exporting beef meat that meets the required market

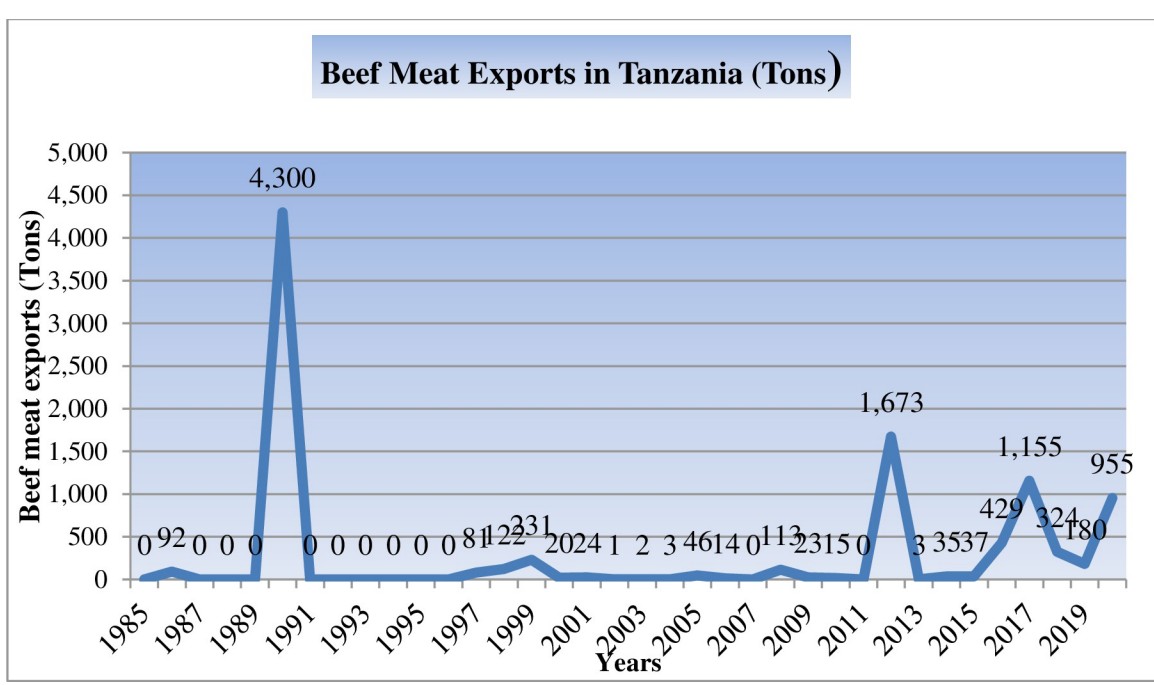

**Fig 2. Trends of beef meat exports in Tanzania from 1985 to 2020 [25].**

specifications of our global customers (global markets) is necessary to guarantee that Tanzanian beef meat producers compete effectively throughout international markets. Meeting global market specifications is also critical in determining the profitability of beef meat-producing farmers (companies) [31]. This is because global customers pay a premium for lines of beef meat that meet specifications (quality standards) while discounting or discarding those that do not [31]. As a result, comprehensive knowledge of the specifications (quality standards) for various global markets allows beef meat producers to suit their possessions, seasonal changes, and inventories with the standards of their target global markets. Regardless of the selected international market, the ability to supply appropriate beef meat supplies consistently benefits the supply chain network, which includes beef meat producers and global customers. As a result, producing high-quality beef meat that meets global market standards is recommended. Tanzanian beef meat producers should indeed maintain their breeding and nourishment programs, as well as strive to enhance their live beef cattle assessment, to ensure that the highest proportion of beef cattle meet the specifications for producing high-quality beef meat for global markets.

**Table 4. General changes and relationships between beef meat exported beef meat output and trade openness from 1985 to 2020.**

| Year | Beef Meat Exported (Tons) | Beef Meat Outputs (Tons) | The Proportion of Beef Meat Exported to Total Beef Meat Output (%) | Trade Openness (%) |
|---|---|---|---|---|
| **1985** | 0 | 162,500 | 0.0 | 7.6 |
| **2020** | 955 | 486,736 | 0.2 | 98.7 |
| **General changes** | 955 (+) | 324,236 (+) | 0.2 (+) | 91.1 (+) |
| **Increase/ decrease** | Increase of 955 tons | Increase of 324,236 tons | Increase the proportion by 0.2% | Increase of 91.1% |

**Source:** FAOSTAT [25], World integrated trade solutions (WITS) [28], and Authors' computations.

## 4.2. Econometric estimation results

**4.2.1. Factors that influenced beef meat exports in Tanzania from 1985 to 2020.** The ordinary least squares (OLS) regression analysis on the factors influencing beef meat exports in Tanzania yielded the R–squared ($R^2$) value of 0.871, while the adjusted $R^2$ was 0.839 as displayed in Table 5. The rest 16.1% is the inexplicable variation in beef meat exports taken by the error term. This suggests that the model's explanatory variables clarify a significant proportion (87.1%) of the variability in Tanzanian beef meat exports. The model was formulated employing SPSS v.22, and it fit well and was statistically significant at *P < 0.05*. As a result, using the estimates in Table 5, the equation to describe Tanzania's beef meat exports ($Iny_1$), relying on its explanatory variables, is expressed mathematically as follows (see Eq 5):

$InBeef\ Meat\ Exports(y_1)$

$$= -0.823 + 0.751 InTerms\ of\ trde - 0.194 InExchange\ rate$$
$$+0.184 InTanzania\ GDP\ per\ capita$$
$$-0.277 InGDP\ per\ capita\ of\ trading\ partners \quad\quad (5)$$
$$+0.059 InGlobal\ beef\ meat\ consumption + 0.574 InTrade\ openness$$
$$+0.181 InBeef\ meat\ outputs + \varepsilon$$

Thus, as shown in Table 5, keeping other factors constant, the terms of trade (TT) had a positive influence on the variability of beef meat export quantity in Tanzania, and it was statically significant at the 5% level of significance. The positive influence implies that if the terms of trade rise by 1%, beef meat export quantity rise by 0.751%. The terms of trade are represented as a ratio of import prices to export prices; that is, the number of imported commodities that an economy can buy for each unit of exported goods/commodities. According to Kumar [32] and Folawewo and Olakojo [33], terms of trade in economics relate to the link between how much a country pays for imports and how much it earns from exports. When the prices of a nation's exports rise above the price of its imports, economists say the terms of trade have improved (positive trade relationship). Any increase in a country's terms of trade is advantageous to the economy since it indicates the country may buy more imports for the same level of exports. As a result, bilateral trade relationships are recommended to boost Tanzanian beef meat exports.

**Table 5. The ordinary least squares (OLS) regression estimates for the factors influencing beef meat exports in Tanzania from 1985 to 2020.**

| Independent Variables in natural log. | Coefficients (β) | Std. Error |
|---|---|---|
| Terms of trade | 0.751* | 0.085 |
| Exchange rates | -0.194* | 0.445 |
| Tanzania GDP per capita | 0.184* | 0.387 |
| GDP per capita of trading partners | -0.277* | 0.320 |
| Global beef meat consumption | 0.059* | 0.409 |
| Trade openness | 0.574* | 0.267 |
| Beef meat outputs | 0.181* | 0.278 |
| Constant | -0.823* | 0.344 |
| R Squared ($R^2$) | 0.871(87.1%) | |
| Adjusted R squared (Adj.$R^2$) | 0.839(83.9%) | |

*Indicate significance level at 5% (P < 0.05). Dependent variable: Beef meat export quantity in Tanzania

The impact of exchange rates on beef meat export markets revealed a negative impact on beef meat exports in Tanzania. Articulating that a one-unit increase in Tanzania Shillings/US$ has a negative influence on beef meat exports of -0.194%. This influence is consistent with the findings of Batten and Belongia [34], Bravo-Ortega and Lederman [35], Lv et al. [36], and Majeed and Ahmad [37]. Inferring that an increase in the power exchange rate will make Tanzanian beef meat less appealing to international customers because more foreign capital will be required to purchase Tanzanian beef meat. The exchange rate is typically the rate at which one country's currency is exchanged for the currency of another country [38]. A steady exchange rate is required for a nation to flourish and keep its economic system stable through exportation (beef meat exports). For instance, if the currency is overvalued and the exchange rate increases versus other currencies, the country's exports become more costly in comparison to the rest of the world, causing beef meat export demand to decline, and vice versa. Moreover, according to the reports by [34, 36, 37], the influence of exchange rates on export volume revealed consistent and highly significant findings of higher export volume as a result of a 1% drop in the exchange rate. This is envisaged when local prices fall owing to exchange rate deflation, making domestic items more competitive to international buyers. Furthermore, some studies imply that both high inflation and devaluation are harmful to the growth of the economy and go on to advise a modest devaluation [39]. Additionally, the exchange rate should be carefully controlled because either appreciation or depreciation has a negative influence on a nation's economy through inhibition of beef meat exportation [40].

The coefficient for GDP per capita in Tanzania had a positive influence on the amount of beef meat exported to international markets and was significant statistically at a 5% level of significance. It indicates that if Tanzania's GDP per capita rises by one US dollar, the amount of beef meat supplied for export rises by 0.184%. Moreover, the Gross domestic product (GDP) has been used as a substitute for a nation's market size. The size of the exporter's markets is anticipated to have a favorable impact on beef meat exports. Tanzania's market size reflects its beef meat export capability. If the value of the coefficient of GDP of the exporting nation is larger than the expected coefficient of GDP of the importing countries (trading partners), the seller may have a domestic impact (enhancing beef meat production) and become a leading exporter [41]. Therefore, because Tanzania's GDP per capita is larger than the GDP per capita of importers (as shown in Table 5), the response of beef meat supply to changes in Tanzania's GDP is higher than the response of demand to changes in importing countries' GDP. This finding shows that Tanzania should look for new markets to widen its beef meat exports. As a result, the Tanzania government should significantly increase its GDP per capita to sustainably boost beef meat exports, which will contribute to the country's economic growth.

Given that Gross domestic product (GDP) has been used as a substitute for a nation's beef meat market size. The size of the importer's markets is anticipated to have a positive impact on beef meat exports. The importers' (trading partners) market size reflects Tanzania's demand for beef meat exports. Nevertheless, the GDP per capita of trading partners had a significant negative impact on the amount of beef meat exported to international markets. Furthermore, the coefficient indicates that a 1% increment in importers' GDP per capita reduces beef meat exports by -0.277%. This is because consumer perceptions of beef meat vary significantly from market to market. For example, a nation's economic growth (GDP) influences typical consumer purchasing powers, with third-world countries emphasizing freshness and safety and wealthy countries emphasizing value and positioning strategies [1]. Furthermore, for Japanese, Korean, Chinese, and United States consumers, the key motivating factors to buy worldwide beef meat include safety, freshness, natural, and value, whereas, for Islamic countries such as Saudi Arabia, Malaysia, and Indonesia, the key motivating factors are halal, freshness, and safety [1]. Given the wide range of Tanzanian beef meat, it is critical to understand

international consumer needs and what pushes buying decisions in each market in promoting beef meat exports.

Moreover, at a 5% ($p < 0.05$) level of significance, the global beef meat consumption factor had a positive influence on the amount of beef meat exported to international markets. This means that, if all other variables remain constant, a one-ton increase in global beef meat consumption has a 0.059% positive impact on Tanzanian beef meat exports. Although beef meat consumption in developed countries (global markets) is also regulated by health perceptions, environmental concerns, and animal welfare concerns to which the exporter must conform, it is inferred to generate demand for beef meat (global markets), hence encouraging beef meat exports among beef meat producers (exporters) including Tanzania [1]. The development and upgrading of food services sectors in emerging markets, as well as increasing consumer attention and consciousness of provenance, sustainability, the welfare of animals, food standards, and reliability, offer additional messaging opportunities for the Tanzania beef meat industry and promote innovative, enterprise initiatives to further differentiation of the beef meat industry. The developing world, notably Asia, has seen an overwhelming beef meat consumption surge over the last decades, with projections for the future decade showing a similar pattern. Surplus output, however, remains concentrated in North and South America, as well as Africa, including Tanzania. Global trade is driven by these production and consumption imbalances [1].

Furthermore, Tanzania being free (with 98.7% of trade openness) to trade with global partners was positively associated with the amount of Tanzanian beef meat exported to the international markets and was statistically significant at the 5% level of significance. The expression of how free or how strictly it is in countries' trade relations with the outside world (trade openness) is believed to enhance trade (exportation) as opposed to trading prohibitions [37, 42]. Several African governments have been concerned about their country's level of openness to foreign trade. According to Yanikkaya [43], trade openness promotes economic growth through benefits gained through exportations. Hence more trade liberalization is recommended in Tanzania to promote global trade and market connectivity, thereby promoting beef meat exports.

Additionally, the assessment evidenced a significantly positive relationship between beef meat exports and beef meat output (production) and was statistically significant at a 5% significance level. This reveals that when beef meat production (output) rises by 1%, beef meat exports are expected to rise by 0.181%. Factors that influence the agribusiness export market (beef meat exports) are reliant on output and the percentage of that output (production) that is distributed between the international and domestic markets [44]. Increased beef meat output leads to an increase in beef meat surplus for exports. As a result, this offers context for the fact that beef meat production is closely related to the success of beef meat exports in Tanzania. As a result, policies and programs to increase beef meat output (production) should be implemented more effectively in Tanzania. One such policy is the National Livestock Policy (NLP), which aims to increase beef meat production, processing, and marketing to meet national nutritional requirements [45]. Furthermore, programs such as the Tanzania Livestock Sector Development Strategy (TLSDS) [15], Tanzania Livestock Modernization Initiative (TLMI) [9], Tanzania Livestock Master Plan (TLMP) [10], Tanzania Development Vision 2025, and Agricultural Sector Development Programme Phase II (ASDP II) all focus on developing the beef meat industry in Tanzania, thereby boosting beef meat exports [46].

## 5. Conclusion

The purpose of this study was to explore the factors (drivers) that influence beef meat exports in Tanzania, particularly focusing on the year 1985 to 2020. The findings revealed that the

elasticities of the terms of trade, Tanzania GDP per capita, global beef meat consumption, trade openness, and beef meat outputs (beef meat production), were found to be positively significant in influencing beef meat exports in Tanzania, while the exchange rate and GDP per capita of trading partners were not. Moreover, the units of these positively influencing factors have been in increasing trends. These results indicate that beef meat exports in Tanzania could increase proportionally; however, beef meat exportation remained minimal. Generally, beef meat exports reached the highest pick of 4,300 tons per year in 1990, whereas from 1991 to date, beef meat exportation in Tanzania has been in declining trends. Nevertheless, while the prospect of Tanzanian beef meat exports appears bright and promising, the industry will continue to encounter trade barriers and must stay competitive (innovative) to produce enough volume and quality beef meat to meet the needs of its existing and expanding markets. This is because Tanzanian beef meat competes for market share with beef meat from other countries in global beef meat markets. Generally, global customers pay a premium for lines of beef meat that meet specifications (quality standards) while discounting or discarding those that do not [31]. As a result, exporting beef meat that meets the required market specifications of our global customers (global markets) is necessary to guarantee that Tanzanian beef meat producers compete effectively throughout international markets. Since, to a large extent, beef meat production in Tanzania is traditional, available data show that 94% of the country's beef meat is predominantly produced under traditional beef meat production, which is known for producing low-quality beef meat [9, 10, 15]. The quantity of beef meat produced has no relevance to its market share in global markets, but its quality standards do. The findings have varying implications as to what factors need to be addressed to further improve beef meat export depending on the stakeholder's point of view, whether it's the farmer or the government. Other factors remaining constant, from the perspective of the farmer (exporter), better access to adequate funds as a result of increased income benefit from export will assist in the production process of beef cattle, either for procuring feeds as well as veterinary inputs, which may lead to enhanced per beef cattle efficiency, resulting in a significant contribution to beef meat exports. Additionally, farmers (exporters) should also keep an eye on changes in exchange rates, which may influence their judgment on how much of their produce to export. Furthermore, from the government's perspective, because trade openness promotes economic growth through export benefits [43], the Tanzanian government and policymakers need to establish balanced policies to strengthen bilateral trade relationships to generate more opportunities in global markets. Nevertheless, given the wide range of Tanzanian beef meat, it is critical to understand international consumer needs and what pushes buying decisions in each market in promoting beef meat exports. Further research should be conducted to determine whether the beef meat produced in Tanzania meets the quality standards of global markets (global consumers). The limited data for 2021 and 2022 hampered current investigations of beef meat exports in Tanzania.

## Supporting information

**S1 Data.**
(SAV)

## Author Contributions

**Conceptualization:** Zhang Yuejie.

**Data curation:** Cornel Anyisile Kibona.

**Formal analysis:** Cornel Anyisile Kibona.

**Funding acquisition:** Zhang Yuejie.

**Investigation:** Cornel Anyisile Kibona.

**Methodology:** Cornel Anyisile Kibona.

**Project administration:** Lu Tian.

**Resources:** Lu Tian.

**Software:** Zhang Yuejie.

**Supervision:** Zhang Yuejie.

**Validation:** Cornel Anyisile Kibona.

**Visualization:** Lu Tian.

**Writing – original draft:** Cornel Anyisile Kibona.

**Writing – review & editing:** Cornel Anyisile Kibona.

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
