## [Decision Letter · Decision Letter 0]

5 Apr 2022

PONE-D-22-05342Towards Developing a Beef Meat Export Oriented Policy in Tanzania: -Exploring the Factors that Influence Beef Meat Exports-PLOS ONE

Dear Dr. Kibona,

Thank you for submitting your manuscript to PLOS ONE. After careful consideration, we feel that it has merit but does not fully meet PLOS ONE’s publication criteria as it currently stands. Therefore, we invite you to submit a revised version of the manuscript that addresses the points raised during the review process. Please submit your revised manuscript by  May 20, 2022. If you will need more time than this to complete your revisions, please reply to this message or contact the journal office at plosone@plos.org. Please include the following items when submitting your revised manuscript:A rebuttal letter that responds to each point raised by the academic editor and reviewer(s). You should upload this letter as a separate file labeled 'Response to Reviewers'.A marked-up copy of your manuscript that highlights changes made to the original version. You should upload this as a separate file labeled 'Revised Manuscript with Track Changes'.An unmarked version of your revised paper without tracked changes. You should upload this as a separate file labeled 'Manuscript'.

We look forward to receiving your revised manuscript.

Kind regards,

László Vasa, PhD

Academic Editor

PLOS ONE

Journal Requirements:

Reviewers' comments:

Reviewer's Responses to Questions

**Comments to the Author**

1. Is the manuscript technically sound, and do the data support the conclusions?

Reviewer #1: Yes

Reviewer #2: Partly

2. Has the statistical analysis been performed appropriately and rigorously? 

Reviewer #1: Yes

Reviewer #2: Yes

3. Have the authors made all data underlying the findings in their manuscript fully available?

Reviewer #1: Yes

Reviewer #2: Yes

4. Is the manuscript presented in an intelligible fashion and written in standard English?

Reviewer #1: Yes

Reviewer #2: Yes

5. Review Comments to the Author

Reviewer #1: The manuscript is technically solid, the data obtained by the authors supports the investigation and its conclusions, and thus the study exhibits novel research findings. What was particularly applauded was the careful examination of the causes for low exports, especially given that with the minor export constraints, Tanzanian beef meat exports could expand proportionately. According to the given data, the authors have not previously published this study. The paper's format is logical, beginning with an overview of the problem and progressing through the methods and sources of data collecting. The article spans the years 1985 through 2020 and employs descriptive and inferential statistics, as well as regression (econometric) analysis models, to examine the data. The studies are carried out to a high technical degree and are sufficiently detailed within the publication, demonstrating that the analysis was carried out appropriately and systematically. The authors have included all data supporting their findings in their publication in tables, along with explanations for those tables. Additionally, the conclusion is given in the appropriate manner, the evidence supports the conclusions, and the policy implications are reasonable and based on the findings. The research complies with all applicable ethical and research integrity standards, and the work adheres to appropriate reporting criteria and community standards for data access. The only (relatively minor) issue with the most recent version of the paper is that it contains a few typos, extra spaces before percentage characters, and missing spaces or letters (e.g. Tanzanian government, p. 10), but my overall recommendation is to publish the paper following these minor revisions to correct the typos.

Reviewer #2: Tha paper main goal is to discover the factors which influence the beef export of Tazania. Zhe topic is somewhat original, however, its statistical approach is, indeed. So I accept it as an original submission.

While I think the manuscript is a good base for a scientific communication, I think the following changes should be made before publication:

- Literature review should be a separate part of the paper. Now it is integrated with the introduction but not enough and isn't made appropriately. So the literature review part should be extended, touching the general issues of the beef value chain and export significance.

- The methodology description is too technical, hard to follow logically - it should be explained, why the selected method was chosen for the research (worth to consider to start the section with the conceptual framework).

- Source for Table 2 is not indicated

- Figure 1 is missing, we just can see the title

- The abstract should be rewritten to be more compact and comprehensive.

- Conclusions are too short and do not contain implications

- Limitations of the research should be indicated

6. PLOS authors have the option to publish the peer review history of their article (what does this mean?). If published, this will include your full peer review and any attached files.

Reviewer #1: No

Reviewer #2: No

---

## [Author Response · Author response to Decision Letter 0]

29 Apr 2022

Response to the Reviewers’ Comments 

We thank the editor and reviewers for taking their time to read and give their comprehensive and constructive comments, which have improved our manuscript. Below; we provide a point by point response to your comments and suggestions and how each one has been addressed in the revision.

Response to Reviewer 1 Comments

Comment 1: The manuscript is technically solid, the data obtained by the authors supports the investigation and its conclusions, and thus the study exhibits novel research findings. What was particularly applauded was the careful examination of the causes for low exports, especially given that with the minor export constraints, Tanzanian beef meat exports could expand proportionately. According to the given data, the authors have not previously published this study. The paper's format is logical, beginning with an overview of the problem and progressing through the methods and sources of data collecting. The article spans the years 1985 through 2020 and employs descriptive and inferential statistics, as well as regression (econometric) analysis models, to examine the data. The studies are carried out to a high technical degree and are sufficiently detailed within the publication, demonstrating that the analysis was carried out appropriately and systematically. The authors have included all data supporting their findings in their publication in tables, along with explanations for those tables. Additionally, the conclusion is given in the appropriate manner, the evidence supports the conclusions, and the policy implications are reasonable and based on the findings. The research complies with all applicable ethical and research integrity standards, and the work adheres to appropriate reporting criteria and community standards for data access. The only (relatively minor) issue with the most recent version of the paper is that it contains a few typos, extra spaces before percentage characters, and missing spaces or letters (e.g. Tanzanian government, p. 10), but my overall recommendation is to publish the paper following these minor revisions to correct the typos.

Response: We thank the reviewer for these comments. We appreciate the compliment from the reviewer that the manuscript is technically solid, the data obtained by the authors supports the investigation and its conclusions, and thus the study exhibits novel research findings. However, few typos, extra spaces before percentage characters, and missing spaces or letters have been corrected.

Response to Reviewer 2 Comments

Comment 1: The paper main goal is to discover the factors which influence the beef export of Tanzania. The topic is somewhat original; however, its statistical approach is, indeed. So I accept it as an original submission.

Response: We thank the reviewer for seeing the useful information contained in our manuscript and accept it as an original submission. 

Comment 2: Literature review should be a separate part of the paper. Now it is integrated with the introduction but not enough and isn't made appropriately. So the literature review part should be extended, touching the general issues of the beef value chain and export significance.

Response: We thank the reviewer for this comment. We are sorry for the shortcomings in our previous manuscript. The literature review part has been extended, touching the general issues of the beef value chain and export significance.

Comment 3: The methodology description is too technical, hard to follow logically - it should be explained, why the selected method was chosen for the research (worth to consider starting the section with the conceptual framework).

Response: We thank the reviewer for this important comment. We have followed the suggestions from the reviewer to indicate why the method was chosen for the research in our current manuscript. Again we have considered starting the section with the conceptual framework.

Comment 4: Source for Table 2 is not indicated.

Response: We thank the reviewer for yet another important reminder. We have indicated the Sources for Table 2.

Comment 5: Figure 1 is missing, we just can see the title

Response: We thank the reviewer for this comment. Figure 1 is included in the manuscript.

Comment 6: The abstract should be rewritten to be more compact and comprehensive?

Response: We thank the reviewer for this comment. We have rewritten the abstract to be more compact and comprehensive. We hope it is clear now.

Comment 7: Conclusions are too short and do not contain implications 

Response: We thank the reviewer for this comment. We double-checked and expanded the conclusions, as well as added implications.

Comment 8: Limitations of the research should be indicated

Response: We thank the reviewer for this comment. We have indicated Limitations of the research in our revised manuscript.

---

## [Decision Letter · Decision Letter 1]

6 Jun 2022

Towards Developing a Beef Meat Export Oriented Policy in Tanzania: -Exploring the Factors that Influence Beef Meat Exports-

PONE-D-22-05342R1

Dear Dr. Kibona,

We’re pleased to inform you that your manuscript has been judged scientifically suitable for publication and will be formally accepted for publication once it meets all outstanding technical requirements.

Kind regards,

László Vasa, PhD

Academic Editor

PLOS ONE

Additional Editor Comments (optional):

Reviewers' comments:

Reviewer's Responses to Questions

**Comments to the Author**

1. If the authors have adequately addressed your comments raised in a previous round of review and you feel that this manuscript is now acceptable for publication, you may indicate that here to bypass the “Comments to the Author” section, enter your conflict of interest statement in the “Confidential to Editor” section, and submit your "Accept" recommendation.

Reviewer #1: All comments have been addressed

2. Is the manuscript technically sound, and do the data support the conclusions?

Reviewer #1: Yes

3. Has the statistical analysis been performed appropriately and rigorously? 

Reviewer #1: Yes

4. Have the authors made all data underlying the findings in their manuscript fully available?

Reviewer #1: Yes

5. Is the manuscript presented in an intelligible fashion and written in standard English?

Reviewer #1: Yes

6. Review Comments to the Author

Reviewer #1: After reading the submission, the adjustments, and the requests made by the other reviewer, I am certain that the authors have responded to all of the changes. Since I was able to see figure one, I believe that the changes have made this submission acceptable.

7. PLOS authors have the option to publish the peer review history of their article (what does this mean?). If published, this will include your full peer review and any attached files.

Reviewer #1: No

---

## [Editor Report · Acceptance letter]

8 Jun 2022

PONE-D-22-05342R1 

Towards Developing a Beef Meat Export Oriented Policy in Tanzania: -Exploring the Factors that Influence Beef Meat Exports- 

Dear Dr. Kibona:

I'm pleased to inform you that your manuscript has been deemed suitable for publication in PLOS ONE. Congratulations! Your manuscript is now with our production department. 

Kind regards, 

on behalf of

Prof. Dr. László Vasa 

Academic Editor

PLOS ONE